# Inherently Fluorescent Peanut-Shaped Polymersomes for Active Cargo Transportation

**DOI:** 10.3390/pharmaceutics15071986

**Published:** 2023-07-20

**Authors:** Jianhong Wang, Yingtong Luo, Hanglong Wu, Shoupeng Cao, Loai K. E. A. Abdelmohsen, Jingxin Shao, Jan C. M. van Hest

**Affiliations:** 1Bio-Organic Chemistry, Institute for Complex Molecular Systems, Eindhoven University of Technology, P.O. Box 513, 5600 MB Eindhoven, The Netherlands; j.wang1@tue.nl (J.W.); y.luo@tue.nl (Y.L.); h.wu@tue.nl (H.W.); l.k.e.a.abdelmohsen@tue.nl (L.K.E.A.A.); 2Max Planck Institute for Polymer Research, 55128 Mainz, Germany; caos@mpip-mainz.mpg.de

**Keywords:** polymersomes, aggregation-induced emission, light-propelled movement, imaging, cargo transportation

## Abstract

Nanomotors have been extensively explored for various applications in nanomedicine, especially in cargo transportation. Motile properties enable them to deliver pharmaceutical ingredients more efficiently to the targeted site. However, it still remains a challenge to design motor systems that are therapeutically active and can also be effectively traced when taken up by cells. Here, we designed a nanomotor with integrated fluorescence and therapeutic potential based on biodegradable polymersomes equipped with aggregation-induced emission (AIE) agents. The AIE segments provided the polymersomes with autofluorescence, facilitating the visualization of cell uptake. Furthermore, the membrane structure enabled the reshaping of the AIE polymersomes into asymmetric, peanut-shaped polymersomes. Upon laser irradiation, these peanut polymersomes not only displayed fluorescence, but also produced reactive oxygen species (ROS). Because of their specific shape, the ROS gradient induced motility in these particles. As ROS is also used for cancer cell treatment, the peanut polymersomes not only acted as delivery vehicles but also as therapeutic agents. As an integrated platform, these peanut polymersomes therefore represent an interesting delivery system with biomedical potential.

## 1. Introduction

Inspired by natural motile systems, over the years, a range of artificial micro/nanomotors have been designed for fulfilling tasks in complex environments, especially in a biomedical context [1,2,3]. These micro/nanomotors are traditionally categorized based on their propulsion mechanism, which is based on either chemical conversion, activation by physical stimuli such as light, or a biological response [4,5,6,7,8,9,10]. To achieve directionality in motility, the propulsion force should be applied anisotropically and /or the particle should possess an asymmetric shape [11]. It has been demonstrated that motile systems achieve efficient transportation and deep tissue penetration [12,13,14]. In this regard, size is important, as micromotors with diameters of more than 5 µm are normally too big for this purpose [15,16,17]. In addition to size, a series of other prerequisites need to be considered when designing nanomotors for applications in the biomedical field [18,19]. First, the building blocks for the preparation of the nanomotors need to be biocompatible and/or biodegradable [20,21]. Second, to visualize and track the nanomotors during transportation, an effective detection signal should be included, which is, in most cases, a fluorescent probe [22,23]. Additionally, when therapeutic agents are transported, their loading efficiency and the prevention of undesired leakage of these compounds should be considered [24].

Aggregation-induced emission (AIE) materials display bright fluorescence when the AIEgens are in an aggregated state. This feature makes them interesting to apply as the detection signal in, for example, the monitoring of cell uptake and biosensing. Furthermore, as AIEgens can also transfer energy from their excited states to other molecules, they have been employed in photodynamic therapy (PDT), where the energy transfer is used to create reactive oxygen species [25,26,27,28]. In recent years, we have demonstrated that AIEgens can be chemically connected to the hydrophobic domain of amphiphilic block copolymers. These polymers were subsequently assembled into spherical polymer vesicles, or polymersomes, while maintaining their AIEgenic features. AIE-based polymersomes have been further developed into nanomotors; for this purpose, spherical AIE-polymersomes were coated with a hemispherical Au shell [29], or cucurbit-shaped and bowl-shaped AIE-polymersomes were decorated or loaded with enzymes to induce autonomous motion [30,31]. Although these nanomotors performed well, in all cases, a multistep procedure had to be followed to include all the necessary elements to enable motile behavior.

In order to construct a multifunctional nanomotor in a more efficient way, we hypothesized that ROS production upon the laser irradiation of AIE polymersomes could lead to motility as long as a product gradient could be created. To achieve this, the particles had to be endowed with an asymmetric topology. Herein, we report the generation of AIEgen functionalized polymersomes with a special peanut-shaped topology that act as nanomotors (Figure 1). The polymersomes were composed of the biodegradable block copolymer poly(ethylene glycol)-poly(trimethylene carbonate) (PEG-PTMC) that was functionalized with aggregation-induced-emission (AIE)-segments. Through a solvent exchange method, the functional block copolymers self-assembled into polymersomes, followed by dialysis against 50 mM NaCl solution to achieve the shape transformation from spheres into peanut-shaped polymersomes. Upon laser irradiation, the polymersomes exhibited autofluorescence and ROS generation [32,33], which indeed triggered autonomous motion. Because of these motile features, the peanut polymersomes could efficiently move to tumorous cells and enhance uptake in a 3D tumor model. At the same time, the stable and bright autofluorescence of the peanut polymersomes enabled the visualization of the whole delivery process. As the functional cargo was structurally integrated with the peanut polymersomes, there were no issues regarding loading efficiency and cargo leaking during transportation. This multifunctional integrated system therefore provides a new platform for efficient therapeutic delivery vehicles.

## 2. Materials and Methods

### 2.1. Materials

All materials were purchased from Sigma-Aldrich without further purification unless stated otherwise. Poly(ethylene glycol) methyl ether 2K was purchased from Rapp Polymers and freeze-dried before use. 1,4-Dioxane and tetrahydrofuran (THF) were obtained from Biosolve Chimie (Metz, France). Dialysis membranes MWCO (12,000–14,000 Da) were supplied by Spectra/Pro^®^ (ThermoFisher, Waltham, MA, USA). Wheat germ agglutinin, Alexa Fluor^TM^ 488 conjugate, calcein-AM, propidium iodide (PI), Dulbecco’s modified eagle medium (DMEM), phosphate-buffered saline (PBS, pH 7.4), no-mycoplasma fetal bovine serum (FBS), trypsin-EDTA and penicillin–streptomycin was purchased from ThermoFisher (Waltham, MA, USA). Ultrapure Milli Q (St. Louis, MI, USA) water (18.2 MΩ·cm) was used in this work.

Information regarding the analytical equipment can be found in the Appendix A.

### 2.2. Synthesis of PEG_44_-P(AIE)_8_

PEG_44_-PTMCP_8_ and amino-functionalized AIE-gen **1** were synthesized according to a previously reported method (Appendix A) [29]. Subsequently, PEG_44_-PTMCP_8_ (92 mg, 0.02 mmoL) was dissolved in 1.5 mL dry THF and cooled to 0 °C. Next, a 0.5 mL THF solution containing AIE-gen **1** (180 mg, 0.18 mmoL) and trimethylamine (34 μL, 0.18 mmoL) was added dropwise. Then, the reaction mixture was stirred for an additional 90 min at room temperature. After the amidation step was completed, the reaction solution was precipitated with diethyl ether three times. Then, the yellow product was isolated and freeze-dried to obtain PEG_44_-P(AIE)_8_. Copolymer composition was calculated by using the protons of PEG (3.61–3.68 ppm), terminal methyl unit (s, 3.38 ppm), TMCAIE CH_2_ (m, 3.70–3.76 ppm), TMCAIE CH_2_ (m, 4.22–4.32 ppm), and TMCAIE CH_2_ (m, 6.60–6.66 ppm). GPC (RI): *M*_n_ (PDI) = 7.94 kDa (1.14).

### 2.3. Peanut Polymersome Preparation

PEG_44_-P(AIE)_8_ (1 mg) was added to 0.5 mL of THF, and the solution was stirred at 750 rpm for an additional dissolution process of 10 min. Then, Milli-Q water (0.5 mL) was injected into the THF solution via a syringe pump at a speed of 0.25 mL/h. The solution turned turbid during the self-assembly process, which indicated the formation of polymersomes. The cloudy solution was transferred into a prehydrated dialysis bag (SpectraPor, MWCO: 12–14 kDa, 2 mL/cm) and dialysis was performed against 50 mM NaCl solution for 24 h with a dialysate change after 1 h. Finally, the peanut polymersomes were obtained and used for subsequent experiments. The physicochemical properties of the peanut polymersomes were characterized by dynamic light scattering (DLS), transmission electron microscopy (TEM), cryogenic transmission electron microscopy (cryo-TEM), ultraviolet–visible spectroscopy (UV-vis), and confocal laser scanning microscopy (CLSM, Leica TCS SP8X).

### 2.4. Spherical AIE-polymersome Preparation

PEG_44_-P(AIE)_8_ (1 mg) was added to 0.5 mL of THF, and the solution was stirred at 750 rpm for an additional dissolution process of 10 min. Then, Milli-Q water (0.5 mL) was injected into the THF solution via a syringe pump at a speed of 0.25 mL/h. The solution turned turbid during the self-assembly process, which indicated the formation of polymersomes. The cloudy solution was transferred into a prehydrated dialysis bag (SpectraPor, MWCO: 12–14 kDa, 2 mL/cm) and dialyzed against Milli-Q water for 24 h with a water change after 1 h. Finally, the polymersomes were obtained and used for subsequent experiments. The morphology and fluorescent properties of the spherical polymersomes were characterized by DLS, TEM, cryo-TEM, and CLSM.

### 2.5. Singlet Oxygen (^1^O_2_) Detection

Singlet oxygen sensor green (SOSG) was employed to evaluate singlet oxygen generation. SOSG was mixed with peanut polymersomes (5 μg/mL, [SOSG] = 10 μM), and the mixture was irradiated with a 405 nm laser (output laser: 1.2 W, 5 min). The generated ^1^O_2_ was determined by recording the SOSG fluorescence intensity with an excitation/emission of 504/525 nm using a microplate reader. The fluorescence intensity of SOSG incubated with peanut polymersomes was normalized to be 100%.

### 2.6. In Vitro Cell Experiments

Cell culture: Human cervical cancer cells (HeLa) and mouse embryonic fibroblast cells (NIH/3T3) were cultured in the cell incubator (Thermo Fisher) in DMEM medium containing 10% FBS, 1% 100 U/mL penicillin, and 1% 100 U/mL streptomycin in 5% CO_2_ at 37 °C and 70% humidity.

Cell viability: To evaluate the feasibility of the peanut polymersomes for nanomedicine applications, their in vitro cytotoxicity was tested with a standard CCK-8 assay. HeLa cells and NIH/3T3 cells were selected as model cell lines. Cells were seeded in 96-well plates at a density of 1 × 10^4^ cells per well with 100 μL of DMEM containing FBS, penicillin, and streptomycin for each well. After incubation overnight, the medium was refreshed, and peanut polymersomes at different concentrations (0, 5, 10, 20, 40, 60, 80, 100, 200 μg/mL) were added. After 24 h incubation, the cells were washed three times with PBS and treated with 100 μL of DMEM containing 10% CCK-8 for each well for 3 h. The absorbance of each well at 450 nm was recorded with a microplate reader. For each group, triplicated wells were tested, and the data were presented as mean ± SD.

^1^O_2_ generation in HeLa cells: The intracellular ^1^O_2_ generation was investigated using the fluorescent probe 2′,7′-dichlorofluorescein diacetate (DCF-DA). HeLa cells were co-incubated with peanut polymersomes (20 μL) for 6 h. The cells were then stained with DCF-DA (20 μM) for 20 min and washed with PBS 3 times. The prepared cells were irradiated with a 405 nm laser (1.2 W, 5 min) and observed with CLSM.

Cell uptake behavior of peanut polymersomes: To evaluate the intracellular uptake behavior, HeLa cells were cultured in μ-Slide 8 wells at a density of 2 × 10^4^ cells per well with 200 μL of DMEM. The cell membrane was stained with Alexa Fluor^TM^ 488. Peanut polymersomes (20 μL) were added to the cell medium immediately before irradiation. For the control group, spherical polymersomes (20 μL) were added to a parallel cell group. To investigate the enhanced cellular uptake behavior owing to the motile properties of peanut polymersomes, the cell membrane was stained with wheat germ agglutinin, Alexa Fluor^TM^ 488 conjugate to show the boundary of the cells. Peanut polymersomes (20 μL) were added to the cell medium and irradiated with a 405 nm laser using CLSM for 5 min followed by co-culturing for 6 h. Then, HeLa cells were characterized by CLSM.

Therapeutic evaluation of peanut polymersomes: To investigate the therapeutic effect of peanut polymersomes, HeLa cells were mixed with peanut polymersomes (50 μg/mL) and immediately irradiated with a laser for 5 min. The parallel experiment in the absence of laser irradiation before incubation was conducted as a control group. After culturing for 6 h, the cells were washed three times with PBS to remove free peanut polymersomes and refreshed with DMEM followed by irradiating with a 405 nm laser for 5 min. After incubation for another 6 h, the cells were stained with calcein-AM/PI and characterized using CLSM.

3D HeLa cellular spheroid penetration of peanut polymersomes: 3D HeLa cellular spheroids were used to further investigate the uptake behavior and tissue penetration of peanut polymersomes. 3D HeLa tumor spheroids were prepared by a “hanging drop” technique according to a previously reported procedure [34]. Briefly, 500 μL of agarose solution (2%, *w*/*v*, PBS) was dropped into the mold of a 3D petri dish. Air bubbles were completely removed via pipette suction. After that, the solidified gels from the mold were placed into each well of a 12-well plate and equilibrated for more than 20 min with DMEM. Then, a cell suspension (190 μL) containing 6 × 10^5^ HeLa cells was slowly added to each well of the 12-well plate, and after standing for 10 min, 2.5 mL of fresh cell culture medium was slowly added. When the tumor spheroids reached the appropriate volume, the penetration capacity and therapeutic efficacy of the peanut polymersomes were evaluated. To study the uptake behavior, the HeLa tumor spheroids were immediately mixed with peanut polymersomes at a volume of 200 μL (peanut polymersomes concentration: 50 μg/mL) and irradiated with a laser. CLSM was used to observe the uptake by applying a standardized Z-stack scanning protocol. Spherical polymersomes as a control group were investigated following the same procedure.

## 3. Results and Discussion

### 3.1. Preparation and Characterization of Peanut Polymersomes

For the preparation of the peanut polymersomes, the amphiphilic block copolymer poly(ethylene glycol)- poly(trimethylene carbonate pentafluoro phenyl ester) was synthesized (Appendix A). Upon amidation with the amine-functional AIE-gen **1**, the polymersome building blocks (PEG_44_-P(AIE)_8_) were attained, which were characterized by ^1^H NMR spectroscopy and GPC (Appendix A) [31]. To form the peanut polymersomes, self-assembly of the PEG_44_-P(AIE)_8_ copolymers was performed using the well-established solvent-induced self-assembly methodology, in which copolymers were dissolved in an organic solvent (THF) at a concentration of 2 mg/mL, after which Milli-Q water (equivalent volume to THF) was added using a syringe pump at a speed of 1 mL/h. Then, the mixture solution was dialyzed against sodium chloride solution (NaCl) (50 mM) to induce the shape transformation from spheres to peanut-shaped polymersomes. The as-prepared peanut polymersomes and spherical polymersomes, which were used as a control group, were characterized using dynamic light scattering (DLS), transmission electron microscopy (TEM), and cryogenic transmission electron microscopy (cryo-TEM) (Figure 1). A significant difference in morphology was observed after dialysis against 50 mM NaCl. Without the NaCl solution, the obtained polymersomes remained as spheres with a diameter of 205 ± 4 nm (Figure 1A–C). Shape transformation into peanut-shaped structures occurred when dialysis against a NaCl solution was performed, leading to particles of 252 ± 4 nm in diameter (Figure 1D–F).

The intrinsic fluorescent properties of AIE-functional polymersomes can be highly beneficial for the imaging and tracking of particles. Consequently, their fluorescence and emission properties were investigated. As shown in Figure 2A,B, peanut polymersomes exhibited strong emission and had maximum absorption in the UV window (λ_Ex_ = 342 nm/λ_Em_ = 601 nm). The fluorescence of both peanut polymersomes and spherical polymersomes was investigated by CLSM (λ_Ex_ = 405 nm/λ_Em_ = 600–700 nm). As shown in Figure 2C,D, both particles exhibited bright fluorescence and could therefore be used for particle tracking and cell uptake studies.

In addition to autofluorescence, the peanut polymersomes could also generate reactive oxygen species (ROS), namely singlet oxygen (^1^O_2_), upon laser irradiation. To evaluate the ^1^O_2_ generation, a fluorescent probe for ^1^O_2_ detection (singlet oxygen sensor green, SOSG) was used. From the change in the SOSG fluorescence intensity, it was demonstrated that peanut polymersomes instantly generated ^1^O_2_ under laser irradiation, whereas the PBS group showed no significant effect (Appendix A). The amount of generated ^1^O_2_ showed a linear relationship with laser irradiation time.

### 3.2. Autonomous Motion

After evaluating the autofluorescence and ROS generation, we investigated the motility of peanut polymersomes upon laser irradiation (405 nm) using nanoparticle tracking analysis (NTA). Compared with the Brownian motion of spherical polymersomes, peanut polymersomes moved faster and exhibited directed motion, which was characterized by the mean square displacement (MSD), diffusion coefficient (D), and motion trajectory (Figure 3A–D). The MSD curves of peanut polymersomes showed a shift from a linear to a parabolic profile once irradiated with UV light (Appendix A). The velocity of each group was calculated from the MSD = (4D)Δt + (V^2^)(Δt^2^), which amounted to ca. 25 ± 3 μm/s for peanut polymersomes and 8 ± 1 μm/s for spherical polymersomes. These results indicate that peanut polymersomes are efficiently propelled. The hypothesized mechanism of the autonomous movement is the asymmetric structure-induced concentration gradient of ^1^O_2_ or related ions, which could explain why spherical polymersomes do not display autonomous motion. Furthermore, we compared the movement performance of peanut polymersomes in both Milli-Q water and biological medium (PBS) (Appendix A).

Peanut polymersomes were slightly slowed down in PBS, which is consistent with previous findings [35]. These results indicate that peanut polymersomes were still motile in a biological environment, which makes them suitable to be used as active delivery vehicles for biomedical applications. To demonstrate the controllable motion, we performed five cycles of turning on and off the incident laser. As shown in Figure 3E, the peanut polymersomes instantly responded by changing their velocity.

### 3.3. Peanut Polymersome-Meditated Cargo Transportation

It has been demonstrated that active delivery systems (micro/nanomotors) can achieve enhanced delivery as well as deep tissue penetration, which improves the therapeutic outcome [36]. We therefore assessed the tissue penetration and cargo delivery of our peanut polymersomes using 2D and 3D cell models. We first evaluated their toxicity toward HeLa cells and NIH/3T3 cells using a CCK-8 assay. Both two cell lines were treated with peanut polymersomes for 24 h. Exposure to peanut polymersomes led to good cell viability (higher than 90%) among all tested concentrations (Appendix A). We subsequently investigated the interaction of peanut polymersomes with HeLa cells using CLSM, benefiting from the particles’ intrinsic fluorescence. To visualize the HeLa cells, the cell membrane was stained with a membrane indicator (wheat germ agglutinin, Alexa Fluor^TM^ 488 conjugate (WGA-AF488)). Then, peanut polymersomes were added to the HeLa cells followed by 5 min of laser irradiation. Spherical polymersomes were used as the control group. As shown in Figure 4, a bright red fluorescence signal originating from the peanut polymersomes was observed on the cell membrane, which means that they quickly accumulated around the HeLa cells. On the contrary, the spherical polymersomes were uniformly distributed in the cell medium under the same conditions. Furthermore, we incubated the peanut polymersomes with HeLa cells for 6 h and treated the samples with and without light irradiation. As shown in Appendix A, the red fluorescence was much higher in the group with light irradiation, indicating that the cellular uptake of peanut polymersomes was enhanced due to their autonomous motion.

Having verified the enhanced cellular uptake using HeLa cells, a 3D tumor model was then used to study tissue penetration. The 3D tumor models were prepared according to a previously reported procedure [34]. Peanut polymersomes and spherical polymersomes were introduced into the tumor culture medium once the diameter of the 3D spheroids reached ca. 300 μm. Then, the 3D spheroids were irradiated with laser light for 5 min using CLSM. Z-scanning imaging sequences (z-stacks) were obtained to observe the distribution of polymersomes at different depths. Compared with the spherical polymersomes, the peanut polymersomes accumulated more on the surface of 3D HeLa spheroids (Appendix A). Spherical polymersomes remained dispersed around the 3D spheroids and did not show any enhanced uptake even when illuminated with light. These results demonstrated that peanut polymersomes can be more efficient than traditional carriers as delivery vehicles due to their motile features.

### 3.4. Photodynamic Therapy Using AIEgenic Peanut Polymersomes

Upon light irradiation, the generation of ROS endows the peanut polymersomes not only with motility but also with a potential therapeutic functionality. The ROS generation from peanut polymersomes was established utilizing the singlet oxygen sensor green fluorescence indicator (Appendix A). Furthermore, to assess ROS production in cells, HeLa cells were incubated with the profluorescent probe 2′,7′-dichlorofluorescein diacetate (DCF-DA) and the peanut polymersomes. DCF-DA emits green fluorescence after cleavage by an intracellular esterase and oxidation via ROS. As shown in Appendix A, strong green fluorescence was observed in HeLa cells, indicating that peanut polymersomes can efficiently convert oxygen to ^1^O_2_. After confirming the ROS generation in HeLa cells, we evaluated the therapeutic effect of peanut polymersomes. First, to achieve active transportation of peanut polymersomes, they were added to HeLa cells and irradiated immediately with a 405 nm laser for 5 min using CLSM. After incubation for 6 h, the free peanut polymersomes were removed, and then the cells were treated with a 405 nm laser irradiation for 5 min to produce ROS intracellularly. Finally, the cells were cultured for another 6 h and stained with calcein-AM and PI to distinguish between live and dead cells, respectively. As shown in Figure 5, dead cells were only detected in the irradiated area, while the non-irradiated cells were still intact. This result shows that peanut polymersomes can be used in photodynamic therapy, with spatial control via the illumination of a specific area. Consequently, peanut polymersomes display integrated functionality with respect to particle visualization, autonomous motion, and photodynamic therapy.

## 4. Conclusions

In summary, we have developed AIE-based polymersomes with peanut-shaped morphology. Due to their asymmetric structure and integrated AIEgenic features, upon laser irradiation, these particles displayed both fluorescence, autonomous motion, and therapeutic potential. The gradient generation of ROS as a result of the asymmetric morphology explains the motile behavior of the particles. Due to this feature, efficient cell uptake and enhanced tissue penetration could be obtained. Furthermore, ROS generation also facilitated photodynamic therapy. Consequently, with this integrated design exploring the inherent AIE properties of these polymersomes, an effective nanomedicine system has been developed, which does not suffer from issues regarding the leakage of therapeutic cargo or fluorescent probes. These AIEgenic asymmetric polymersomes therefore provide a new approach toward the use of active transporters in nanomedicine.

## Data Availability

The data presented in this work are available on request from the corresponding authors.

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
