# Peer review of "Inherently Fluorescent Peanut-Shaped Polymersomes for Active Cargo Transportation"

_pharmaceutics, 2023, doi:10.3390/pharmaceutics15071986_

Round 1
Reviewer 1 Report
The manuscript is of high interest and practical importance. The original nanomotor with integrated fluorescence and therapeutic potential has been developed using biodegradable polymersomes equipped with aggregation-induced emission agents. Peanut-shaped polymersomes show also ability to produce reactive oxygen species allowing their application as chemotherapeutics.
The manuscript is well-prepared and logically presented. The experimental data are meaningful. I have concern regarding toxicity of the synthesized polymersomes towards normal cells. This point is out of consideration in current work.
The second point is the absence of the file with supplementary materials.
After clarification of these points, the manuscript can be accepted to publication.
Reviewer 2 Report
The manuscript by Jianhong Wang et al. designed motor systems that are therapeutically active and can also be effectively traced when taken up by cells. The authors have provided interesting results and I have the following suggestions:
1, the authors should put the Figure S10 and Figure S11 in the main text. These are vey important resutls to show the therapeutic effects of peanut polymersomes.
2, the authors must further discuss the limitations of the current study. The authors have not tested the therapeutic effects of peanut polymersomes in vivo.
3, Have the authors checked the therapeutic effects of peanut polymersomes in other cancer cells? The authors must explain.
Round 2
Reviewer 1 Report
The revised version of the manuscript is improved vs. initial submission. The requred toxicity test for normal cells is performed. The supplementary materials are provided and correctly reflect the manuscript content. The answers to the comments are meaningful.
The manuscript can be accepted to publication as is.
Reviewer 2 Report
The authors have revised the manuscript accordingly. It can be considered for publication.